# End-to-end Task-oriented Dialogue:
# A Survey of Tasks, Methods, and Future Directions

**Libo Qin[1], Wenbo Pan[2], Qiguang Chen[3], Lizi Liao[4], Zhou Yu[5], Yue Zhang[6]**
**Wanxiang Che[3], Min Li[1]**

[1]School of Computer Science and Engineering, Central South University
[2]Harbin Institute of Technology, China
[3]Research Center for Social Computing and Information Retrieval
[4]Singapore Management University
[5]Department of Computer Science, Columbia University
[6]School of Engineering, Westlake University
{lbqin, min.li}@csu.edu.cn, pixelwenbo@gmail.com,
{qgchen, car}@ir.hit.edu.cn, lzliao@smu.edu.sg,
zy2461@columbia.edu, yue.zhang@wias.org.cn

## Abstract

End-to-end task-oriented dialogue (EToD) can directly generate responses in an end-to-end fashion without modular training, which attracts escalating popularity. The advancement of deep neural networks, especially the successful use of large pre-trained models, has further led to significant progress in EToD research in recent years. In this paper, we present a thorough review and provide a unified perspective to summarize existing approaches as well as recent trends to advance the development of EToD research. The contributions of this paper can be summarized: (1) *First survey*: to our knowledge, we take the first step to present a thorough survey of this research field; (2) *New taxonomy*: we first introduce a unified perspective for EToD, including (i) *Modularly EToD* and (ii) *Fully EToD*; (3) *New Frontiers*: we discuss some potential frontier areas as well as the corresponding challenges, hoping to spur breakthrough research in EToD field; (4) *Abundant resources*: we build a public website[1], where EToD researchers could directly access the recent progress. We hope this work can serve as a thorough reference for the EToD research community.

## 1 Introduction

Task-oriented dialogue systems (ToD) can assist users in achieving particular goals with natural language interaction such as booking a restaurant or navigation inquiry. This area is seeing growing interest in both academic research and indus-

---

[1]We collect the related papers, baseline projects, and leaderboards for the community at https://etods.net/.

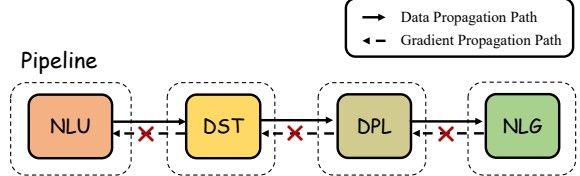

(a) Traditional pipeline task-oriented dialogue framework.

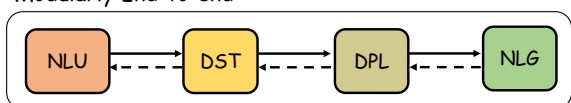

(b) Modularly end-to-end task-oriented dialogue framework.

Unified Sequence-to-sequence Model

(c) Fully end-to-end task-oriented dialogue framework.

Figure 1: Pipeline Task-oriented Dialogue System (a), Modularly End-to-end Task-oriented Dialogue System (b) and Fully End-to-end Task-oriented Dialogue System (c). The dashed box denotes separately trained while the solid line box represents end-to-end training.

try deployment. As shown in Figure 1(a), conventional ToD systems utilize a pipeline approach that includes four connected modular components: (1) natural language understanding (NLU) for extracting the intent and key slots of users (Qin et al., 2020a, 2021b); (2) dialogue state tracking (DST) for tracing users' belief state given dialogue history (Balaraman et al., 2021a; Jacqmin et al., 2022a); (3) dialogue policy learning (DPL) to determine the next step to take (Kwan et al., 2022); (4) natural language generation (NLG) for generating dialogue system response (Wen et al., 2015; Li et al., 2020).

While impressive results have been achieved in

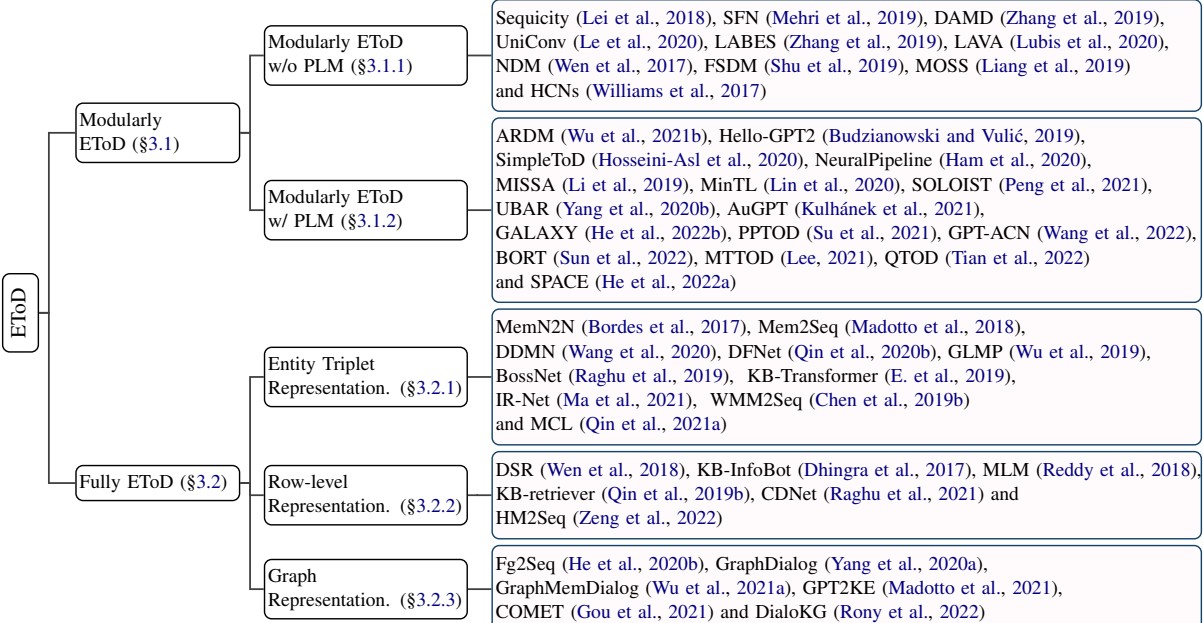

Figure 2: Taxonomy for End-to-end Task-orient Dialogue (EToD).

previous pipeline ToD approaches, they still suffer from two major drawbacks. (1) Since each module (*i.e.,* NLU, DST, DPL, and NLG) is trained separately, pipeline ToD approaches cannot leverage shared knowledge across all modules; (2) As the pipeline ToD solves all sub-tasks in sequential order, the errors accumulated from the previous module are propagated to the latter module, resulting in an error propagation problem. To solve these issues, dominant models in the literature shift to end-to-end task-oriented dialogue (EToD). A critical difference between traditional pipeline ToD and EToD methods is that the latter can train a neural model for all the four components simultaneously (see Fig. 1(b)) or directly generate the system response via a unified sequence-to-sequence framework (see Fig. 1(c)).

Thanks to the advances of deep learning approaches and the evolution of pre-trained models, recent years have witnessed remarkable success in EToD research. However, despite its success, there remains a lack of a comprehensive review of recent approaches and trends. To bridge this gap, we make the first attempt to present a survey of this research field. According to whether the intermediate supervision is required and KB retrieval is differentiable or not, we provide a unified taxonomy of recent works including (1) modularly EToD (Mehri et al., 2019; Le et al., 2020) and (2) fully EToD (Eric and Manning, 2017; Wu et al., 2019; Qin et al., 2020b). Such taxonomy can cover all types of EToD , which help researchers to track the progress of EToD comprehensively. Furthermore, we present some potential future directions and summarize the challenges, hoping to provide new insights and facilitate follow-up research in the EToD field.

Our contributions can be summarized as follows:

(1) **First survey:** To our knowledge, we are the first to present a comprehensive survey for end-to-end task-oriented dialogue system;

(2) **New taxonomy:** We introduce a new taxonomy for EToD including (1) *modularly EToD* and (2) *fully EToD* (as shown in Fig. 2);

(3) **New frontiers:** We discuss some new frontiers and summarize their challenges, which shed light on further research;

(4) **Abundant resources:** we make the first attempt to organize EToD resources including open-source implementations, corpora, and paper lists at https://etods.net/.

We hope that this work can serve as quick access to existing works and motivate future research[2].

## 2  Background

This section describes the definition of modularly end-to-end task-oriented dialogue (Modularly

---

[2]Due to the page limitation, the detailed related work section can be found in the Appendix B.

EToD §2.1) and fully end-to-end task-oriented dialogue (Fully EToD §2.2), respectively.

## 2.1 Modularly EToD

Modularly EToD typically generates system response through sub-components (e.g., dialog state tracking (DST), dialogue policy learning (DPL) and natural language generation NLG)). Unlike traditional ToD which trains each component (*e.g.,* DST, DPL, NLG) separately, modularly EToD trains all components in an end-to-end manner where the parameters of all components are optimized simultaneously.

Formally, each dialogue turn consists of a user utterance $u$ and system utterance $s$. For the $n$-th dialog turn, the agent observes the dialogue history $\mathcal{H} = (u_1, s_1), (u_2, s_2), ..., (u_{n-1}, s_{n-1}), u_n$ and the corresponding knowledge base (KB) as $\mathcal{KB}$ while it aims to predict a system response $s_n$, denoted as $\mathcal{S}$.

Modularly EToD first reads the dialogue history $\mathcal{H}$ to generate a belief state $\mathcal{B}$:

$$\mathcal{B} = \text{Modularly\_EToD}(\mathcal{H}), \quad (1)$$

where $\mathcal{B}$ consists of various slot value pairs (e.g., price: cheap) for each domain.

The generated belief state $\mathcal{B}$ is used to query the corresponding $\mathcal{KB}$ to obtain the database query results $\mathcal{D}$:

$$\mathcal{D} = \text{Modularly\_EToD}(\mathcal{B}, \mathcal{KB}), \quad (2)$$

Then, $\mathcal{H}$, $\mathcal{B}$, and $\mathcal{D}$ is used to decide dialogue action $\mathcal{A}$. Finally, modularly EToD generates the final dialogue system response $\mathcal{S}$ conditioning on $\mathcal{H}$, $\mathcal{B}$, $\mathcal{D}$ and $\mathcal{A}$:

$$\mathcal{S} = \text{Modularly\_EToD}(\mathcal{H}, \mathcal{B}, \mathcal{D}, \mathcal{A}), \quad (3)$$

## 2.2 Fully End-to-end Task-oriented Dialogue

In comparison to modularly EToD, Fully EToD (Eric and Manning, 2017) has two crucial differences: (1) modularly EToD leverages the generated beliefs as API to query KB, which is non-differentiable. In contrast, fully EToD directly encodes KB and uses a neural network to query the KB in a differentiable manner. (2) Unlike modularly EToD which requires modular annotation (*e.g.,* DST, DPL annotation) for intermediate supervision, fully EToD can directly generate system response given only dialogue history and the corresponding KB;

Formally, fully EToD can be denoted as:

$$\mathcal{S} = \text{Fully\_EToD}(\mathcal{H}, \mathcal{KB}). \quad (4)$$

## 3 Taxonomy of EToD Research

This section describes the progress of EToD according to the new taxonomy including modularly EToD (§3.1) and Fully EToD (§3.2).

## 3.1 Modularly EToD

We further divide the modularly EToD into two sub-categories (1) modularly EToD without a pre-trained model (§3.1.1) and (2) modularly EToD with a pre-trained model (§3.1.2) according to whether or not a pre-trained model is used, which are shown in Fig. 3 (a) and (b).

### 3.1.1 Modularly EToD without PLM

One line of work mainly focuses on optimizing the whole dialogue with supervised learning (SL) while another line considers incorporating a reinforcement learning (RL) approach for optimizing.

**Supervised Learning.** Liu and Lane (2017) first presented an LSTM-based (Hochreiter and Schmidhuber, 1997) model which jointly learns belief tracking and KB retrieval. Wen et al. (2017) also proposed an EToD model with a modularized design, in which each module transmits its latent representation instead of predicted labels to the next module. Lei et al. (2018) introduced `Sequicity`, a two-stage `CopyNet` (Gu et al., 2016), merging belief tracking and response generation in a sequence-to-sequence model. `MOSS` (Liang et al., 2019) expanded `Sequicity` with NLU and DPL modules for comprehensive dialogue supervision. Shu et al. (2019) modeled language understanding and state tracking tasks jointly using a unified seq2seq approach and separate GRUs for different slot types. Mehri et al. (2019) explicitly incorporated the dialogue structure information into EToD, enhancing the domain generalizability. Zhang et al. (2019) considered multiple appropriate responses under the same context in ToD and improved dialogue policy diversity by balancing the valid output action distribution. `LABES` (Zhang et al., 2020b) leveraged unlabeled dialogue data (*i.e.,* without belief state labels) to achieve semi-supervised learning of ToD.

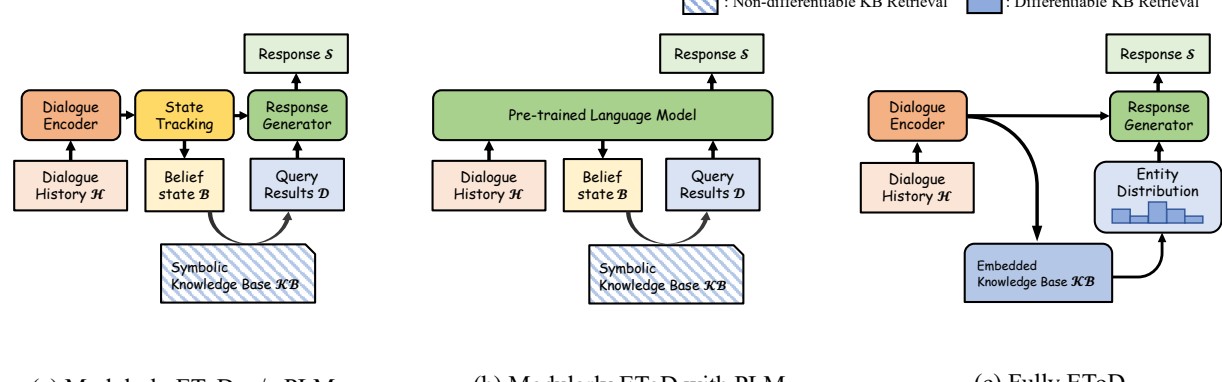

Figure 3: Three categories for EToD, including (a) Modularly EToD without PLM; (b) Modularly EToD with PLM and (c) Fully EToD. Modularly EToD generates the system response with modularized components and train all components in an end-to-end fashion (see (a) and (b)). Meanwhile, the KB retrieval of modularly EToD is by API call that is non-differentiable. In contrast, fully EToD can directly generate system response given the dialogue history and KB, which does not require the modularized components (see (c)). Besides, the KB retrieval process in fully EToD is differentiable and can be optimized together with other parameters in EToD.

**Reinforcement Learning.** Reinforcement Learning (RL) has been explored as a supplement to supervised learning in dialogue policies optimization. Li et al. (2018) demonstrated less error propagation using RL-optimized networks than SL settings. SL signals have also been incorporated into RL frameworks, either by modifying rewards (Zhao and Eskenazi, 2016) or adding SL cycles (Liu et al., 2017). Approaches like LAVA (Lubis et al., 2020), LaRL (Zhao et al., 2019), CoGen (Ye et al., 2022) and HDNO (Wang et al., 2021) have explored the modeling of latent representations. Work on RL-optimized EToD training with human intervention includes HCNs (Williams et al., 2017), human-corrected model predictions (Liu et al., 2018; Liu and Lane, 2018), and determining optimal time for human intervention (Rajendran et al., 2019; Wang et al., 2019).

### 3.1.2 Modularly End-to-end Task-oriented Dialogue with Pre-trained Model

There are two main streams of PLM for modularly EToD including (1) Decoder-only PLM (Radford et al.) and (2) Encoder-Decoder PLM (Lewis et al., 2019; Raffel et al., 2020).

**Decoder-only PLM.** Some works adopted GPT-2 (Radford et al.) as the backbone of EToD models. Budzianowski and Vulić (2019) first attempted to employ a pretrained GPT model for EToD, which considers dialogue context, belief state, and database state as raw text input for the GPT model to generate the final system response. Wu et al.

(2021b) introduced two separate GPT-2 models to learn the user and system utterance distribution effectively. Hosseini-Asl et al. (2020) proposed SimpleToD, recasting all ToD subtasks as a single sequence prediction paradigm by optimizing for all tasks in an end-to-end manner. Wang et al. (2022) re-formulated the task-oriented dialogue system as a natural language generation task. UBAR (Yang et al., 2020b) followed the similar paradigm with SimpleTOD. The core difference is that UBAR incorporated all belief states in all dialogue turns while SimpleToD only utilized belief states of the last turn.

Another series of works tried to modify the pre-training objective of autoregressive transformers. To this end, Li et al. (2019) replaced system response ground truth with random distractor at a possibility during training and leveraged a next utterance classifier to distinguish them. Soloist (Peng et al., 2021) proposed an auxiliary task where the target belief state is replaced with the belief state from unrelated samples for consistency prediction. Kulhánek et al. (2021) further augmented GPT-2 by presenting a new dialogue consistency classification task. The experimental results show that these more challenging training objectives bring significant improvements.

**Encoder-decoder PLM.** PLMs with an encoder-decoder architecture such as BART (Lewis et al., 2019), T5 (Raffel et al., 2020) and UniLM (Dong et al., 2019) are also explored in

| Model | MultiWOZ2.0 | | | | MultiWOZ2.1 | | | |
|---|---|---|---|---|---|---|---|---|
| | Inform (%) | Success (%) | BLEU | Combined | Inform (%) | Success (%) | BLEU | Combined |
| *Modularly End-to-end Task-oriented Dialogue without Pre-trained Model* | | | | | | | | |
| MD-Sequicity (Lei et al., 2018) | - | - | - | - | 66.4 | 45.3 | 15.5 | 71.4 |
| SFN+RL (Mehri et al., 2019) | 73.8 | 58.6 | 16.9 | 83.0 | 73.8 | 58.6 | 16.9 | 83.0 |
| DAMD (Zhang et al., 2019) | 76.3 | 60.4 | 16.6 | 85.0 | 76.4 | 60.4 | 16.6 | 85.0 |
| UniConv (Le et al., 2020) | - | - | - | - | 72.6 | 62.9 | 19.8 | 87.6 |
| LABES (Zhang et al., 2020b) | - | - | - | - | 78.1 | 67.1 | 18.1 | 90.7 |
| LAVA (Lubis et al., 2020) | 91.8 | 81.8 | 12.0 | 98.8 | - | - | - | - |
| *Modularly End-to-end Task-oriented Dialogue with Pre-trained Model* | | | | | | | | |
| SimpleToD (Hosseini-Asl et al., 2020) | 84.4 | 70.1 | 15.0 | 92.3 | 85.0 | 70.5 | 15.2 | 93.0 |
| UBAR (Yang et al., 2020b) | 95.4 | 80.7 | 17.0 | 105.1 | 95.7 | 81.8 | 16.5 | 105.3 |
| MinTL-BART (Lin et al., 2020) | 84.9 | 74.9 | 17.9 | 97.8 | - | - | - | - |
| AuGPT (Kulhánek et al., 2021) | 83.1 | 70.1 | 17.2 | 93.8 | 83.5 | 67.3 | 17.2 | 92.6 |
| SOLOIST (Peng et al., 2021) | 85.5 | 72.9 | 16.5 | 95.7 | 85.5 | 72.9 | 16.5 | 95.7 |
| MTTOD (Lee, 2021) | 91.0 | 82.6 | 21.6 | 108.3 | 91.0 | 82.1 | 21.0 | 107.5 |
| PPTOD (Su et al., 2021) | 89.2 | 79.4 | 18.6 | 102.9 | 87.1 | 79.1 | 19.2 | 102.3 |
| SimpleToD-ACN (Wang et al., 2022) | 85.8 | 72.1 | 15.5 | 94.5 | - | - | - | - |
| GALAXY (He et al., 2022b) | 94.4 | 85.3 | 20.5 | 110.4 | 95.3 | 86.2 | 20.0 | 110.8 |
| SPACE3 (He et al., 2022a) | 95.3 | 88.0 | 19.3 | 111.0 | 95.6 | 86.1 | 19.9 | 110.8 |
| BORT (Sun et al., 2022) | 93.8 | 85.8 | 18.5 | 108.3 | - | - | - | - |

Table 1: Modularly EToD performance on MultiWOZ2.0 and MultiWOZ2.1. The highest scores are marked with underlines. We adopted reported results from published literature (Zhang et al., 2020b, 2019; He et al., 2022b).

| Model | Match | Success | BLEU |
|---|---|---|---|
| *Modularly EToD without Pre-trained Model* | | | |
| NDM (Wen et al., 2017) | 90.4 | 83.2 | 21.2 |
| Sequicity (Lei et al., 2018) | 92.7 | 85.4 | 25.3 |
| FSDM (Shu et al., 2019) | 93.5 | 86.2 | 25.8 |
| MOSS (Liang et al., 2019) | 95.1 | 86.0 | 25.9 |
| LABES-S2S (Zhang et al., 2020b) | 96.4 | 82.3 | 25.6 |
| *Modularly EToD with Pre-trained Model* | | | |
| ARDM (Wu et al., 2021b) | - | 86.2 | 25.4 |
| SOLOIST (Peng et al., 2021) | - | 87.1 | 25.5 |
| BORT (Sun et al., 2022) | - | 89.7 | 25.9 |
| SPACE3 (He et al., 2022a) | 97.7 | 88.2 | 23.7 |

Table 2: Modularly EToD performance on CamRest676 (Wen et al., 2017). We adopted reported results from published literature (Zhang et al. (2020b); Sun et al. (2022)). Match metric measures whether the entity chosen at the end of each dialogue aligns with the entities specified by the user.

EToD. MinTL (Lin et al., 2020) considered training EToD with PLMs in the Seq2Seq manner, where two different decoders are introduced to track belief state and predict response, respectively. PPToD (Su et al., 2021) recast ToD subtasks into prompts and leveraged the multitask transfer learning of T5 (Raffel et al., 2020). Huang et al. (2022) embedded KB information into the language model for implicit knowledge access.

In addition, another series of works devised unique pre-training objectives for encoder-decoder transformers. GALAXY (He et al., 2022b) introduced a dialog act prediction pre-training task for policy optimization. Godel (Peng et al., 2022) leveraged a new phase of grounded pre-training

designed to improve adaptation ability. BORT (Sun et al., 2022) added a denoising reconstruction task to reconstruct the original context from generated dialogue states. MTTOD (Lee, 2021) introduced a span prediction pre-training task. SPACE-3 (He et al., 2022a) further improved over GALAXY with UniLM backbone, where five pre-training objectives are applied to better understand semantic information for task-oriented dialogue. Recently, encoder-decoder PLMs have shown the potential of converting EToD into other task forms like QA (Tian et al., 2022; Xie et al., 2022).

### 3.1.3 Leaderboard and Takeaway.

**Leaderboard:** Leaderboard for the widely used datasets: MultiWOZ2.0, MultiWOZ2.1 and Camrest676 is shown in Table 1 and Table 2. The widely used metrics are BLEU, Inform, Success, and Combined. Detailed descriptions of datasets and metrics are shown in Appendix A.1.

**Takeaway:** As seen, we have the following observations: (1) *PLM Attains Improvement.* We observe that most modularly EToD with PLM outperforms the modularly EToD without PLM, which indicates that knowledge inferred from a pre-trained model can benefit EToD; (2) *Shared Knowledge Leverage.* Since each module (*i.e.,* NLU, DST, PL, NLG) is highly related, modularly EToD can enable the model to fully utilize shared knowledge across all modules.

| Embedding Technique | Related Work | Illustration |
|---|---|---|
| a. Entity Triplet Representation | MemN2N (Bordes et al., 2017) , KVRet (Eric and Manning, 2017), Mem2Seq (Madotto et al., 2018), BossNet (Raghu et al., 2019), GLMP (Wu et al., 2019), DDMN (Wang et al., 2020), DFNet (Qin et al., 2020b), IR-Net (Ma et al., 2021), WMM2Seq (Chen et al., 2019b), MCL (Qin et al., 2021a) |  |
| b. Row-level Representation | KB-InfoBot (Dhingra et al., 2017), MLM (Reddy et al., 2018), CDNet (Raghu et al., 2021), DSR (Wen et al., 2018), KB-Retriever (Qin et al., 2019b), HM2Seq (Zeng et al., 2022) |  |
| c. Graph Representation | GraphDialog (Yang et al., 2020a), Fg2seq (He et al., 2020b), DialoKG (Rony et al., 2022), GraphMemDialog (Wu et al., 2021a), COMET (Gou et al., 2021), MAKER (Wan et al., 2023) |  |

Table 3: Three types of KB Representation in EToD, including (a) entity triple representation; (b) row-level representation and (c) graph representation.

## 3.2 Fully EToD

In the following, we describe the recent dominant fully EToD works according to the category of KB representation, which is illustrated in Fig. 3(c).

### 3.2.1 Triplet Representation.

Specifically, given a knowledge base (KB), triplet representation stores each KB entity in a (*subject, relation, object*) representation. For example, all triplets can be formularized as (*centric entity of $i^{th}$ row, slot title of $j^{th}$ column, entity of $i^{th}$ row in $j^{th}$ column*). (*e.g.,* (Valero, Type, Gas Station)).

The KB entity representation is calculated by the sum of the word embedding of the subject and relation using bag-of-words approaches. It is one of the most widely used approaches for representing KB. Specifically, Eric and Manning (2017) employed a key-value retrieval mechanism to retrieve KB knowledge triplets. Other works treat KB and dialogue history equally as triplet memories (Madotto et al., 2018; Wu et al., 2019; Chen et al., 2019b; He et al., 2020a; Qin et al., 2021a). Memory networks (Sukhbaatar et al., 2015) have been applied to model the dependency between related entity triplets in KB (Bordes et al., 2017; Wang et al., 2020) and improves domain scalability (Qin et al., 2020b; Ma et al., 2021). To improve the response quality with triplet KB representation, Raghu et al. (2019) proposed BOSS-NET to disentangle NLG and KB retrieval and Hong et al. (2020) generated responses through a template-filling decoder.

### 3.2.2 Row-level Representation.

While triplet representation is a direct approach for representing KB entities, it has the drawback of ignoring the relationship across entities in the same row. To migrate this issue, some works investigated the row-level representation for KB.

In particular, KB-InfoBot (Dhingra et al., 2017) first utilized posterior distribution over KB rows. Reddy et al. (2018) proposed a three-step retrieval model, which can select relevant KB rows in the first step. Wen et al. (2018) used entity similarity as the criterion for selecting relevant KB rows. Qin et al. (2019b) employed a two-step retrieving procedure by first selecting relevant KB rows and then choosing the relevant KB column. Recently, Zeng et al. (2022) proposed to store KB rows and dialogue history into two separate memories.

### 3.2.3 Graph Representation

Though row-level representation achieves promising performance, they neglect the correlation between KB and dialogue history. To solve this issue, a series of works focus on better contextualizing entity embedding in KB by densely connecting entities and corresponding slot titles in dialogue history. This can be done with either graph-based reasoning or attention mechanism, where entity presentations are fully aware of other entities or dialogue context. To this end, Yang et al. (2020a) facilitated

| | SMD | | | | | MultiWOZ2.1 | | | | |
|---|---|---|---|---|---|---|---|---|---|---|
| Model | BLEU | Ent.F1(%) | Sch.F1(%) | Wea.F1(%) | Nav.F1(%) | BLEU | Ent.F1(%) | Res.F1(%) | Att.F1(%) | Hot.F1(%) |
| *Entity Triplet Representation* | | | | | | | | | | |
| KVRet (Eric and Manning, 2017) | 13.2 | 48.0 | 62.9 | 53.3 | 44.5 | - | - | - | - | - |
| Mem2Seq (Madotto et al., 2018) | 12.6 | 33.4 | 49.3 | 32.8 | 20.0 | 6.6 | 21.6 | 22.4 | 22.0 | 21.0 |
| GLMP (Wu et al., 2019) | 14.8 | 60.0 | 69.6 | 62.6 | 53.0 | 6.9 | 32.4 | 38.4 | 24.4 | 28.8 |
| BossNet (Raghu et al., 2019) | 8.3 | 35.9 | 50.2 | 34.5 | 21.6 | 5.7 | 25.3 | 26.2 | 24.8 | 23.4 |
| KB-Transformer (E. et al., 2019) | 13.9 | 37.1 | 51.2 | 48.2 | 23.3 | - | - | - | - | - |
| DDMN (Wang et al., 2020) | 17.7 | 55.6 | 65.0 | 58.7 | 47.2 | 12.4 | 31.4 | 30.6 | 32.9 | 30.6 |
| DFNet (Qin et al., 2020b) | 14.4 | 62.7 | 73.1 | 57.6 | 57.9 | 9.4 | 35.1 | 40.9 | 28.1 | 30.6 |
| TToS (He et al., 2020a) | 17.4 | 55.4 | 63.5 | 64.1 | 45.9 | - | - | - | - | - |
| IR-Net (Ma et al., 2021) | 16.3 | 63.2 | - | - | - | 10.9 | 37.5 | - | - | - |
| MCL (Qin et al., 2021a) | 17.2 | 60.9 | 70.6 | 62.6 | 59.0 | - | - | - | - | - |
| *Row-level Representation* | | | | | | | | | | |
| DSR (Wen et al., 2018) | 12.7 | 51.9 | 52.1 | 50.4 | 52.0 | 9.1 | 30.0 | 33.4 | 28.0 | 27.1 |
| MLM (Reddy et al., 2018) | 15.6 | 55.5 | 67.4 | 54.8 | 45.1 | 9.2 | 27.8 | 29.8 | 27.4 | 25.2 |
| KB-retriever (Qin et al., 2019b) | 13.9 | 53.7 | 55.6 | 52.2 | 54.5 | - | - | - | - | - |
| HM2Seq (Zeng et al., 2022) | 14.6 | 63.1 | 73.9 | 64.4 | 56.2 | - | - | - | - | - |
| *Graph Representation* | | | | | | | | | | |
| Fg2Seq (He et al., 2020b) | 16.8 | 61.1 | 73.3 | 57.4 | 56.1 | 13.5 | 36.0 | 40.4 | 41.7 | 30.9 |
| GraphDialog (Yang et al., 2020a) | 13.7 | 60.7 | 72.8 | 55.2 | 54.2 | - | - | - | - | - |
| GraphMemDialog (Wu et al., 2021a) | 18.8 | 64.5 | 75.9 | 62.3 | 56.3 | 14.9 | 40.2 | 42.8 | 48.8 | 36.4 |
| GPT2+KE (Madotto et al., 2021) | 17.4 | 63.1 | 72.6 | 57.7 | 53.5 | - | - | - | - | - |
| COMET (Gou et al., 2021) | 17.3 | 63.6 | 77.6 | 58.3 | 56.0 | - | - | - | - | - |
| Modularized Pre-Training (Qin et al., 2023b) | 18.8 | 63.8 | 75.0 | 58.4 | 59.1 | 13.6 | 36.3 | 41.5 | 36.2 | 31.2 |
| DialoKG (Rony et al., 2022) | 20.0 | 65.9 | - | - | - | - | - | - | - | - |
| UnifiedSKG (Xie et al., 2022) | - | 67.9 | - | - | - | - | - | - | - | - |
| MAKER (Wan et al., 2023) | 25.9 | 71.3 | - | - | - | 18.8 | 54.7 | - | - | - |

Table 4: Fully EToD performance on SMD and MultiWOZ2.1. Ent.F1, Sch.F1, Wea.F1, Nav.F1, Res.F1, Att F1.and Hot.F1 stand for the abbreviation of Entity F1, Schedule F1, Weather F1, Navigation F1, Restaurant F1 and Hotel F1, respectively. We adopted reported results from published literature (Qin et al., 2020b; Wu et al., 2021a; Wang et al., 2020; Gou et al., 2021)

entity contextualization by applying graph-based multi-hop reasoning on the entity graph. Wu et al. (2021a) proposed a graph-based memory network to yield context-aware representations. Another series of works leveraged transformer architecture to learn better entity representation, where the dependencies between dialogue history and KB were learned via self-attention (He et al., 2020b; Gou et al., 2021; Rony et al., 2022; Qin et al., 2023b; Wan et al., 2023).

### 3.2.4 Leaderboard and Takeaway

**Leaderboard:** A comprehensive leaderboard for the widely used dataset: SMD and Multi-WOZ2.1 is shown in Table 4. The widely used metrics for fully EToD are BLEU and F1. Detailed information of datasets and metrics are shown in Appendix A.2.

**Takeaway:** Compaunderline to modular EToD, fully EToD brings two major advantages. (1) ***Human Annotation Efforts Underlineuction.*** Modularly EToD still requires modular annotation data for intermediate supervision. In contrast, fully EToD only requires the dialogue-response pairs, which can greatly underlineuce human annotation efforts; (2) ***KB Retrieval End-to-end Training.*** Unlike the non-differentiable KB retrieval in modularly EToD, fully EToD can optimize the KB retrieval process in a fully end-to-end paradigm, which can enhance the KB retrieval ability.

## 4 Future Directions

This section will discuss new frontiers for EToD, hoping to facilitate follow-up research in this field.

### 4.1 LLM for EToD

Recently, Large Language Models (LLMs) have gained considerable attention for their impressive performance across various Natural Language Processing (NLP) benchmarks (Touvron et al., 2023; OpenAI, 2023; Driess et al., 2023). These models are capable to execute predetermined instructions and interface with external resources, such as APIs (Patil et al., 2023) and knowledge databases. This positions LLMs as promising candidates for end-to-end dialogue systems (EToD). Existing research has also explored to apply LLMs in task-oriented dialogue (ToD) scenarios, using both few-shot and zero-shot learning paradigms (Pan et al., 2023; Heck et al., 2023; Hudevcek and Dusek, 2023; Parikh et al., 2023).

However, several critical challenges remain to be addressed in EToD in future research. We summarize the main challenges as follows:

1. **Safety and Risk Mitigation:** LLMs like chatbots can sometimes generate harmful or biased responses (OpenAI, 2023), posing serious safety concerns. It is crucial to improve their controllability and interpretability. One promising approach is integrating human feed-

back during training (Bai et al., 2022; Chung et al., 2022).

2. **Complex Conversations Management:** LLMs have limitations in managing complex, multi-turn dialogues (Heck et al., 2023; Pan et al., 2023). EToDs often require advanced context modeling and reasoning abilities, which is an area ripe for improvement.

3. **Domain Adaptation:** For task-oriented dialogue, LLMs need to gain specific domain knowledge. However, simply suppling knowledge with finetuning or prompting may lead to problems like catastrophic forgetting or biased attention (Liu et al., 2023). Finding a balanced approach for knowledge adaptation remains a challenge.

In addition to these challenges, there are also emerging opportunities that could further enhance the capabilities of LLMs in EToD systems. These opportunities are summarized below:

1. **Meta-learning & Personalization:** LLMs can adapt quickly with limited examples. This paves the way for personalized dialogues through meta-learning algorithms.

2. **Multi-agent Collaboration & Self-learning from Interactions:** The strong language modeling capabilities of LLMs make self-learning from real-world user interactions more feasible (Park et al., 2023). This can advance collaborative, task-solving dialogue agents

## 4.2 Multi-KB Settings

Recent EToD models are limited to single-KB settings where a dialogue is supported by a single KB, which is far from the real-world scenario. Therefore, endowing EToD with the ability of reasoning over multiple KBs for each dialogue plays a vital role in a real-world deployment. To this end, Qin et al. (2023a) take the first meaningful step to the multi-KB EToD.

The main challenges for multi-KB settings are as follows: (1) `Multiple KBs Reasoning`: How to reason across multiple KBs to retrieve relevant knowledge entries for dialogue generation is a unique challenge; (2) `KB Scalibility`: When the number of KBs becomes larger in real-world scenarios, how to effectively represent all the KBs in a single model is non-trivial.

## 4.3 Pre-training Paradigm for Fully EToD

Pre-trained Language Models have shown remarkable success in open-domain dialogues. ((Bao et al., 2021; Shuster et al., 2022)). However, there is relatively little research addressing how to pre-train a fully EToD. We argue that the main reason for hindering the development of pre-training fully EToD is the lack of large amounts of knowledge-grounded dialogue for pre-training.

We summarize the core challenges for pre-training fully EToD: (1) `Data Scarcity`: Since the annotated KB-grounded dialogues are scarce, how to automatically augment a large amount of training data is a promising direction; (2) `Task-specific Pre-training`: Unlike the traditional general-purpose mask language modeling pre-training objective, the unique challenge for a task-oriented dialogue system is how to make KB retrieval. Therefore, how to inject KB retrieval ability in the pre-training stage is worth exploring.

## 4.4 Knowledge Transfer

With the development of traditional pipeline task-oriented dialogue systems, there exist various powerful modularized ToD models, such as NLU (Qin et al., 2019a; Zhang et al., 2020a), DST (Dai et al., 2021; Guo et al., 2022; Chen et al., 2022), DPL (Chen et al., 2019a; Kwan et al., 2022) and NLG (Wen et al., 2015; Li et al., 2020). A natural and interesting research question is how to transfer the dialogue knowledge from well-trained modularized ToD models to modularly or fully EToD.

The main challenge for knowledge transfer is `Knowledge Preservation`: How to balance the knowledge learned from previous modularized dialogue models and current data is an interesting direction to explore.

## 4.5 Reasoning Interpretability

Current fully EToD models perform knowledge base (KB) retrieval via a differentiable attention mechanism. While appealing, such a black-box retrieval method makes it difficult to analyze the process of KB retrieval, which can seriously hurt the user's trust. Inspired by Wei et al. (2022); Zhang et al. (2022), employing a chain of thought in KB reasoning in fully EToD is a promising direction to improve the interpretability of KB retrieval.

The main challenge for the direction is `design of reasoning steps:` how to propose an ap-

propriate chain of thought (e.g., when to retrieve rows and when to retrieve columns) to express the KB retrieval process is non-trivial.

## 4.6 Cross-lingual EToD

Current success heavily relies on large amounts of annotated data that is only available for high-resource language (*i.e.,* English), which makes it difficult to scale to other low-resource languages. Actually, with the acceleration of globalization, task-oriented dialogue systems like Google Home and Apple Siri are required to serve a diverse user base worldwide, across various languages, which cannot be achieved by the previous monolingual dialogue. Therefore, zero-shot cross-lingual direction that can transfer knowledge from high-resource language to low-resource languages is a promising direction to solve the problem. To this end, Lin et al. (2021) and Ding et al. (2022) introduced BiToD and GlobalWoZ benchmarks to promote cross-lingual task-oriented dialogue.

The main challenge for zero-shot cross-lingual EToD includes: (1) `Knowledge base Alignment:` A unique challenge for cross-lingual EToD is the knowledge base (KB) alignment. How to effectively align the KB structure information across different languages is an interesting research question to investigate; (2) `Unified Cross-lingual Model:` Since different modules (e.g., DST, DPL, and NLG) have heterogeneous structural information, how to build a unified cross-lingual model to align dialogue information across heterogeneous input in all languages is a challenge.

## 4.7 Multi-modal EToD

Current dialogue systems mainly handle plain text input. Actually, we experience the world with multiple modalities (*e.g.,* language and image). Therefore, building a multi-modal EToD system that is able to handle multiple modalities is an important direction to investigate. Unlike the traditional single-modal dialogue system which can be supported by the corresponding KB, multi-modal EToD requires both the KB and image features to yield an appropriate response.

The main challenges for multi-modal EToD are as follows: (1) `Multimodal Feature Alignment and Complementary:` How to effectively make a multimodal feature alignment and complementary to better understand the dialogue is a crucial ability for multi-modal EToD; (2)

`Benchmark Scale Limited:` Current multimodal dataset such as MMConv (Liao et al., 2021) and SIMMC 2.0 (Kottur et al., 2021) are slightly limited in size and diversity, which hinders the development of multi-modal EToD. Therefore, building a large benchmark plays a vital role for promoting multi-modal EToD.

## 5 Conclusion

We made a first attempt to summarize the progress of end-to-end task-oriented dialogue systems (EToD) by introducing a new perspective of recent work, including modularly EToD and fully EToD. In addition, we discussed some new trends as well as their challenges in this research field, hoping to attract more breakthroughs on future research.

## Acknowledgements

This work was supported by the National Natural Science Foundation of China (NSFC) via grant 62306342, 62236004 and 61976072 and sponsored by CCF-Baidu Open Fund. This work was also supported by the Science and Technology innovation Program of Hunan Province under Grant No. 2021RC4008. We are grateful for resources from the High Performance Computing Center of Central South University.

## Limitation

This study presented a comprehensive review and unified perspective on existing approaches and recent trends in end-to-end task-oriented dialogue systems (EToD). We have also created the first public resources website to help researchers stay updated on the progress of EToD. However, the current version primarily focuses on high-level comparisons of different approaches, such as overall system performance, rather than a fine-grained analysis. In the future, we intend to include more in-depth comparative analyses to gain a better understanding of the advantages and disadvantages of various models, such as comparing KB retrieval results and performance across different domains.

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

## A  Datasets and Metrics

### A.1  Datasets and Metrics for Modularly EToD

#### A.1.1  Dataset

Three commonly used datasets for modularly EToD are `CamRest676`, `MultiWOZ2.0`, and `MultiWOZ2.1`.

**CamRest676**   (Wen et al., 2017) is a relatively small-scale restaurant domain dataset, which consists of 408/136/136 dialogues for training/validation/test.

**MultiWOZ2.0**   (Budzianowski et al., 2018) is one of the most widely used ToD dataset. It contains over 8,000 dialogue sessions and 7 different domains including: restaurant, hotel, attraction, taxi, train, hospital and police domain.

**MultiWOZ2.1**   (Eric et al., 2019) is an improved version of `MultiWOZ2.0`, where incorrect slot annotations and dialogue acts were fixed.

#### A.1.2  Metrics

The widely used metrics for modularly EToD are `BLEU`, `Inform`, `Success`, and `Combined`.

**BLEU**   (Papineni et al., 2002) is used to measure the fluency of generated response by calculating n-gram overlaps between the generated response and the gold response.

**Inform and Success**   (Budzianowski et al., 2018). `Inform` measures whether the system provides an appropriate entity and `Success` measures whether the system answers all requested attributes.

**Combined**   (Budzianowski et al., 2018) is a comprehensive metric considering `BLEU`, `Inform`, and `Success`, which can be calculated by: Combined = (Inform + Success ) × 0.5 + BLEU).

### A.2  Datasets and Metrics for Fully EToD

#### A.2.1  Dataset

`SMD` (Eric and Manning, 2017) and `MultiWOZ2.1` (Qin et al., 2020b) are two popular datasets for evaluating fully EToD.

**SMD**   Eric and Manning (2017) proposed a Stanford Multi-turn Multi-domain Task-oriented Dialogue Dataset, which includes three domains: navigation, weather, and calendar.

**MultiWOZ2.1.**   Qin et al. (2020b) introduces an extension of `MultiWOZ2.1` where they annotate the corresponding KB for each dialogue.

#### A.2.2  Metrics

Fully EToD adopts `BLEU` and `Entity F1` to evaluate the fluent generation and KB retrieval ability, respectively.

**BLEU**   has been described in Section A.1.1.

**Entity F1**   Eric and Manning (2017) is used to measure the difference between entities in the system and gold responses by micro-averaging the precision and recall.

## B  Related Work

Modular task-oriented dialogues typically consist of spoken language understanding (SLU), dialogue state tracking (DST), dialogue manager (DM) and natural language generation (NLG), which have achieved significant success. Recently, numerous surveys summaries the recent progress of modular task-oriented dialogue systems. Specifically, Louvan and Magnini (2020); Larson and Leach (2022) and Qin et al. (2021c) summarize the recent progress of neural-based models for SLU. On DST, Balaraman et al. (2021b) and Jacqmin et al. (2022b) review the recent neural approaches and highlight the need for greater exploration on generalizability within the field. In terms of dialogue management, Dai et al. (2020) concentrates on challenges like model scalability, data scarcity, and improving training efficiency. For natural language generation (NLG), Santhanam and Shaikh (2019) provides a comprehensive overview of the past, present, and future directions of NLG. Finally, Chen et al. (2017), Zhang et al. (2020c) and Ni et al. (2023) provide an overarching review of the dialogue system as a whole, emphasising the impact of deep learning technologies.

Compared to the existing work, we focus on the end-to-end task-oriented dialogue system. To the best of our knowledge, this is the first comprehensive survey of the end-to-end task-oriented dialogue system. We hope that this survey can attract more breakthroughs on future research.