# OpenReview forum: "End-to-end Task-oriented Dialogue: A Survey of Tasks, Methods, and Future Directions"
_EMNLP/2023/Conference — EMNLP 2023 Main_

### Official Review · Reviewer_334w · 2023-08-01

**Soundness:** 3

**Excitement:**

3: Ambivalent: It has merits (e.g., it reports state-of-the-art results, the idea is nice), but there are key weaknesses (e.g., it describes incremental work), and it can significantly benefit from another round of revision. However, I won't object to accepting it if my co-reviewers champion it.

**Missing References:**

Q-TOD: A Query-driven Task-oriented Dialogue System

Multi-Grained Knowledge Retrieval for End-to-End Task-Oriented Dialog, are not included in the paper.

**Paper Topic And Main Contributions:**

The paper presents a comprehensive review and introduces a unified perspective on existing EToD approaches and recent trends. It aims to advance the development of EToD research by highlighting its contributions, including being the survey in the field,  discussing potential frontier areas and challenges, and providing abundant resources through a public website for EToD researchers. Overall, the paper contributes to the EToD research community by presenting a  survey and introducing a new taxonomy.
Strengths:
The paper provides a thorough overview of the progress made in the field of task-oriented dialogue and examines emerging frontier areas while summarizing the associated challenges.
The writing is accessible and straightforward, making the content easily understandable to readers.
Weaknesses:
Lack of Discussion on Integration with Large Pre-trained Models: In the era of large pre-trained models, the paper fails to address the integration of task-oriented dialogue with these models, which could have significant implications for EToD research.
Omissions of Recent Progress: Some noteworthy recent developments in EToD, such as QToD (Q-TOD: A Query-driven Task-oriented Dialogue System) and Multi-Grained Knowledge Retrieval for End-to-End Task-Oriented Dialog, are not included in the paper.

**Reasons To Accept:**

The paper provides a thorough overview of the progress made in the field of task-oriented dialogue and examines emerging frontier areas while summarizing the associated challenges.

The writing is accessible and straightforward, making the content easily understandable to readers.

**Reasons To Reject:**

Lack of Discussion on Integration with Large Pre-trained Models: In the era of large pre-trained models, the paper fails to address the integration of task-oriented dialogue with these models, which could have significant implications for EToD research.

Omissions of Recent Progress: Some noteworthy recent developments in EToD, such as QToD (Q-TOD: A Query-driven Task-oriented Dialogue System) and Multi-Grained Knowledge Retrieval for End-to-End Task-Oriented Dialog, are not included in the paper.

**Reproducibility:**

4: Could mostly reproduce the results, but there may be some variation because of sample variance or minor variations in their interpretation of the protocol or method.

**Reviewer Confidence:**

4: Quite sure. I tried to check the important points carefully. It's unlikely, though conceivable, that I missed something that should affect my ratings.

---

> ### Author Rebuttal · Authors · 2023-08-28
>
> Thanks for your acknowledgments and interest on our work! We notice your only main concerns are the (1) ***Lack of Discussion on Integration with Large Pre-trained Models*** and (2) ***Omissions of Recent Progress***. However, we sincerely think there might be some misconceptions. In the following, **we will clarify your concerns and we would greatly appreciate it if you can re-consider the work considering our clarification**.
>
> **Question1**: Lack of Discussion on Integration with Large Pre-trained Models.
>
> **Answer1**: Thank you for your valuable comment. We sincerely think that the main contribution of our work is making the first attempt to provide a comprehensive survey for end-to-end task-oriented dialogue, including new taxonomy, new frontiers and available resources, which are acknowledged by you, Reviewer ynBS and Reviewer TKmZ. Adding discussion on integration with large pre-trained models will help to enrich our work and we can easily provide more analysis in the Future Directions section in the next version.
>
> We will follow your suggestion to add a detailed discussion on integration with large pre-trained models for end-to-end task-oriented dialogue in the next version. For example, some challenges and opportunities on task-oriented dialogue systems brought by LLMs are listed as follows:
>
> **Challenges**:
>
> 1. Safety and Risk Mitigation: As highlighted in the GPT4 technical report [1], LLMs like chatbots can sometimes generate harmful or biased responses, posing serious safety concerns. It is crucial to improve their controllability and interpretability. One promising approach is integrating human feedback during training [2].
>
> 2. Complex Conversations: LLMs still have limitations in managing complex, multi-turn dialogues [3]. These often require advanced context modeling and reasoning abilities, which is an area ripe for improvement.
>
> 3. Domain Knowledge Acquisition: For task-oriented dialogue, LLMs need to gain specialist knowledge. However, brutal-force domain knowledge injection may lead to problems like catastrophic forgetting or biased attention [4]. Finding a balanced approach to knowledge acquisition remains a challenge.
>
> **Opportunities**:
>
> 1. Multimodal Ability: Recent advancements have enabled the pairing of LLMs with vision models [5,6]. This presents an opportunity to create richer, multimodal dialogues, as LLMs can process visual signals in addition to text.
>
> 2. Meta-learning & Personalization: LLMs have shown a remarkable ability to adapt quickly from a few examples, as noted in a recent survey [7]. Utilizing meta-learning algorithms, LLMs could offer personalized dialogue based on user preferences.
>
> 3. Multi-agent Collaboration & Self-learning from interactions: LLMs' strong language modeling makes self-learning from real user interactions more feasible [8]. This could emulate real-world collaborative task-solving and elevate the level of dialogue agent intelligence.
>
> [1] OpenAI. GPT-4 Technical Report. ArXiv abs/2303.08774.
>
> [2] Anthropic. Claude: an AI assistant focused on being helpful, harmless, and honest.
>
> [3] Pan et al. A Preliminary Evaluation of ChatGPT for Zero-shot Dialogue Understanding. ArXiv abs/2304.04256.
>
> [4] Liu et al. Lost in the Middle: How Language Models Use Long Contexts. ArXiv abs/2307.03172.
>
> [5] Driess et al. PaLM-E: An Embodied Multimodal Language Model. Presented at the International Conference on Machine Learning.
>
> [6] Peng et al. Kosmos-2: Grounding Multimodal Large Language Models to the World. ArXiv abs/2306.14824.
>
> [7] Zhao et al. A Survey of Large Language Models. ArXiv abs/2303.18223.
>
> [8] Park et al. Generative Agents: Interactive Simulacra of Human Behavior. ArXiv abs/2304.03442.
>
> **Question 2**: Omissions of Recent Progress: Some noteworthy recent developments in EToD, such as QToD (Q-TOD: A Query-driven Task-oriented Dialogue System) and Multi-Grained Knowledge Retrieval for End-to-End Task-Oriented Dialog, are not included in the paper.
>
> **Answer 2**: Thanks for your kind mention. **We sincerely believe this is a misunderstanding**. Actually, the mentioned work (1) Q-TOD: A Query-driven Task-oriented Dialogue System (Q-ToD) [1] and (2) Multi-Grained Knowledge Retrieval for End-to-End Task-Oriented Dialog (MAKER) [2] have been included in our paper.
>
> For the QToD work, we categorized it under Modularly EToD w/ PLM in Figure 2. Moreover, in section 3.1.2, we specifically highlighted and discussed its innovative paradigm of shaping the ToD task into QA formats. This represents our understanding and appreciation of its significance in task-oriented dialogue systems.
>
> For the MAKER paper, we grouped it under Graph Representation in Table 3. We also highlighted MAKER's latest SOTA results on two ToD leaderboards in Table 4. Finally, in section 3.2.3, we have further discussed the general characteristics of such works, demonstrating our deep understanding of its value and relevance to the field.
>
> We will add more discussion in the next version.
>
> [1] Xin Tian, Yingzhan Lin, Mengfei Song, Fan Wang, Huang He, Shuqi Sun, and Hua Wu. 2022. Q-TOD: A Query-driven Task-oriented Dialogue System.
>
> [2] Fanqi Wan, Weizhou Shen, Ke Yang, Xiaojun Quan, and Wei Bi. 2023. Multi-grained knowledge retrieval for end-to-end task-oriented dialog.

---

### Official Review · Reviewer_TKmZ · 2023-08-03

**Typos Grammar Style And Presentation Improvements:** The last sentence of the title in Fig…
**Soundness:** 4

**Excitement:**

3: Ambivalent: It has merits (e.g., it reports state-of-the-art results, the idea is nice), but there are key weaknesses (e.g., it describes incremental work), and it can significantly benefit from another round of revision. However, I won't object to accepting it if my co-reviewers champion it.

**Paper Topic And Main Contributions:**

This paper is well-written, with clear and attractive formatting, and strong readability. The literature review on end-to-end task-oriented dialogue is detailed, and the summary and analysis are comprehensive.

One shortcoming is that the Future Directions section does not discuss the opportunities and challenges brought by large language models to this field.

**Reasons To Accept:**

Please refer to the “Paper Topic And Main Contributions” part.

**Reasons To Reject:**

Please refer to the “Paper Topic And Main Contributions” part.

**Reproducibility:**

4: Could mostly reproduce the results, but there may be some variation because of sample variance or minor variations in their interpretation of the protocol or method.

**Reviewer Confidence:**

4: Quite sure. I tried to check the important points carefully. It's unlikely, though conceivable, that I missed something that should affect my ratings.

---

> ### Author Rebuttal · Authors · 2023-08-28
>
> Thanks for your acknowledgements and interests on our work. We will carefully address your questions as below:
>
> **Question1**: Future Directions section should discuss the opportunities and challenges brought by large language models to this field.
>
> **Answer1**: Thank you for your insightful feedback. This is a very good suggestion to improve our work. We will follow your suggestion to add the opportunities and challenges brought by large language models (LLM) for end-to-end task-oriented dialogue system in the next version. For example, some challenges and opportunities on task-oriented dialogue systems brought by LLMs are listed as follows:
>
> **Challenges**:
>
> 1.	Safety and Risk Mitigation: As highlighted in the GPT4 technical report [1], LLMs like chatbots can sometimes generate harmful or biased responses, posing serious safety concerns. It is crucial to improve their controllability and interpretability. One promising approach is integrating human feedback during training [2].
>
> 2.	Complex Conversations: LLMs still have limitations in managing complex, multi-turn dialogues [3]. These often require advanced context modeling and reasoning abilities, which is an area ripe for improvement.
>
> 3.	Domain Knowledge Acquisition: For task-oriented dialogue, LLMs need to gain specialist knowledge. However, brutal-force domain knowledge injection may lead to problems like catastrophic forgetting or biased attention [4]. Finding a balanced approach to knowledge acquisition remains a challenge.
>
> **Opportunities**:
>
> 1.	Multimodal Ability: Recent advancements have enabled the pairing of LLMs with vision models [5,6]. This presents an opportunity to create richer, multimodal dialogues, as LLMs can process visual signals in addition to text.
>
> 2.	Meta-learning & Personalization: LLMs have shown a remarkable ability to adapt quickly from a few examples, as noted in a recent survey [7]. Utilizing meta-learning algorithms, LLMs could offer personalized dialogue based on user preferences.
>
> 3.	Multi-agent Collaboration & Self-learning from interactions: LLMs' strong language modeling makes self-learning from real user interactions more feasible [8]. This could emulate real-world collaborative task-solving and elevate the level of dialogue agent intelligence.
>
>
> [1] OpenAI. GPT-4 Technical Report. ArXiv abs/2303.08774.
>
> [2] Anthropic. Claude: an AI assistant focused on being helpful, harmless, and honest.
>
> [3] Pan et al. A Preliminary Evaluation of ChatGPT for Zero-shot Dialogue Understanding. ArXiv abs/2304.04256.
>
> [4] Liu et al. Lost in the Middle: How Language Models Use Long Contexts. ArXiv abs/2307.03172.
>
> [5] Driess et al. PaLM-E: An Embodied Multimodal Language Model. Presented at the International Conference on Machine Learning.
>
> [6] Peng et al. Kosmos-2: Grounding Multimodal Large Language Models to the World. ArXiv abs/2306.14824.
>
> [7] Zhao et al. A Survey of Large Language Models. ArXiv abs/2303.18223.
>
> [8] Park et al. Generative Agents: Interactive Simulacra of Human Behavior. ArXiv abs/2304.03442.

---

### Official Review · Reviewer_ynBS · 2023-08-05

**Soundness:** 4

**Excitement:**

4: Strong: This paper deepens the understanding of some phenomenon or lowers the barriers to an existing research direction.

**Paper Topic And Main Contributions:**

Goal:  To systematically survey end-to-end task oriented dialog (TOD) systems

Details
1. The paper is the first survey of end-to-end TOD systems.
2. The taxonomy (modular e2e v/s fully e2e) is clear and important.
3. Authors provide a public website containing all resources which can be beneficial for the TOD community.
4. Leaderboards and collation of results can help progress of the field

**Questions For The Authors:**

Have authors done a survey of the recent evaluation protocols for modular and full end-to-end TOD systems?

**Reasons To Accept:**

This work is important as it is the first survey of the field. The paper has clear taxonomy, good coverage, leaderboards and good discussion on future directions. All the relevant data is made publicly available. Overall, this work can provide visibility, transparency and traction to the field.

**Reasons To Reject:**

The paper lacks detailed discussion on the evaluation protocols for end-to-end TOD. This includes automatic metrics, human evaluation and recent LLM based metrics like BERTScore and its variant.

**Reproducibility:**

4: Could mostly reproduce the results, but there may be some variation because of sample variance or minor variations in their interpretation of the protocol or method.

**Reviewer Confidence:**

4: Quite sure. I tried to check the important points carefully. It's unlikely, though conceivable, that I missed something that should affect my ratings.

---

> ### Author Rebuttal · Authors · 2023-08-28
>
> Thanks for your acknowledgements and interests on our work. We will carefully address your question as below:
>
> **Question1**: adding detailed discussion on the evaluation protocols for end-to-end TOD is better. This includes automatic metrics, human evaluation and recent LLM based metrics like BERTScore and its variant.
>
> **Answer1**: Thank you for your constructive comments. We totally agree that a detailed discussion on the evaluation protocols for end-to-end TOD is important. To this end, we have summarized the widely used automatic evaluation metrics for both modular and full end-to-end TOD systems, which is illustrated in the appendix due to the page limitation. We will follow your suggestions to add more discussions on human evaluation and recent LLM based metrics in the next version.

---

### Meta-Review · Area_Chair_nLJD · 2023-09-11

**Recommendation:** 4

**Metareview:**

This paper presents a survey on end-to-end task-oriented dialogue systems. The survey stands out due to the following features:

1. It is the first survey to comprehensively summarize efforts in this field.
2. It introduces a new taxonomy: modular ETOD and Fully ETOD.
3. The inclusion of resources such as a public website and leadership board.
4. A discussion on frontier trends in the field.

The soundness scores were recorded as (4, 4, 3). All reviewers found the paper well-written and easy to follow. The retrospection and analysis of prior work are both detailed and comprehensive. The fresh taxonomy is lucid, and the provision of resources like a public website and leadership board can potentially enhance visibility, transparency, and traction in the domain.

The excitement scores were (4, 3, 3). Despite its strengths, the work has a few areas of improvement:

1. The challenges associated with evaluations are not sufficiently addressed.
2. While Large Language Models (LLMs) are gaining prominence in NLP research, their significance isn't thoroughly discussed.

In conclusion, this work is **Sound and Moderately Exciting**.

---

### Decision · Program_Chairs · 2023-10-07

**Decision:**

Accept-Main

**Comment:**

This paper presents a survey on end-to-end task-oriented dialogue systems. The survey stands out due to the following features:

1. It is the first survey to comprehensively summarize efforts in this field.
2. It introduces a new taxonomy: modular ETOD and Fully ETOD.
3. The inclusion of resources such as a public website and leadership board.
4. A discussion on frontier trends in the field.

The soundness scores were recorded as (4, 4, 3). All reviewers found the paper well-written and easy to follow. The retrospection and analysis of prior work are both detailed and comprehensive. The fresh taxonomy is lucid, and the provision of resources like a public website and leadership board can potentially enhance visibility, transparency, and traction in the domain.

The excitement scores were (4, 3, 3). Despite its strengths, the work has a few areas of improvement:

1. The challenges associated with evaluations are not sufficiently addressed.
2. While Large Language Models (LLMs) are gaining prominence in NLP research, their significance isn't thoroughly discussed.

In conclusion, this work is **Sound and Moderately Exciting**.